# Integrative mRNA and microRNA Analysis Exploring the Inducing Effect and Mechanism of Diallyl Trisulfide (DATS) on Potato against Late Blight

**DOI:** 10.3390/ijms24043474

**Published:** 2023-02-09

**Authors:** Yongfei Jian, Shun Feng, Airong Huang, Zhiming Zhu, Jiaomei Zhang, Shicai Tang, Liang Jin, Maozhi Ren, Pan Dong

**Affiliations:** 1School of Life Sciences, Chongqing University, Chongqing 401331, China; 2Chongqing Key Laboratory of Biology and Genetic Breeding for Tuber and Root Crops, Chongqing 400716, China; 3Sanya Nanfan Research Institute, School of Horticulture, Hainan University, Haikou 570228, China; 4Hongshen Honors School, Chongqing University, Chongqing 401331, China; 5Institute of Urban Agriculture, Chinese Academy of Agricultural Sciences, Chengdu 610213, China

**Keywords:** oomycete disease, *Phytophthora infestans*, biocontrol, transcriptome

## Abstract

Potato late blight, caused by *Phytophthora infestans*, leads to a significant reduction in the yield and value of potato. Biocontrol displays great potential in the suppression of plant diseases. Diallyl trisulfide (DATS) is a well-known natural compound for biocontrol, although there is little information about it against potato late blight. In this study, DATS was found to be able to inhibit the hyphae growth of *P. infestans*, reduce its pathogenicity on detached potato leaves and tubers, and induce the overall resistance of potato tubers. DATS significantly increases catalase (CAT) activity of potato tubers, and it does not affect the levels of peroxidase (POD), superoxide dismutase (SOD), and malondialdehyde (MDA). The transcriptome datasets show that totals of 607 and 60 significantly differentially expressed genes (DEGs) and miRNAs (DEMs) are detected. Twenty-one negatively regulated miRNA-mRNA interaction pairs are observed in the co-expression regulatory network, which are mainly enriched in metabolic pathways, biosynthesis of secondary metabolites, and starch and sucrose metabolism based on the KEGG pathway. Our observations provide new insight into the role of DATS in biocontrol of potato late blight.

## 1. Introduction

As the world’s fourth largest food crop, potato (*Solanum tuberosum* L.) is important for food security in the world [1]. Late blight is the most devastating disease in the potato production process and during storage, causing potato yield and quality loss [2]. The Great Irish Famine of the 1870s resulted from potato late blight [3]. Infection of potato tubers occurs directly through the spread of its phytopathogen *P. infestans* zoospores in the field and indirectly by contacting the infected tubers during storage [4]. Tubers with *P. infestans* will cause the disease to transmit to previously uninfected potato production areas if used as seeds [5]. Some traditional chemical fungicides (mancozeb, infinito, phosphorous acid, azoxystrobin, and hydrogen peroxide, etc.) were used in a large amount in the field and storage warehouse for controlling the late blight [6]. However, the abuse of chemical fungicides, serious environmental pollution, considerable residues of agricultural products, and chemical resistance of pathogens pose a serious threat to human health and environmental safety [7,8]. Selection of biological control agents (such as plant-derived active ingredients, microbial antagonistic bacteria, etc.) as substitutes for chemical fungicides is of great significance to the sustainable development of agricultural ecology.

Allicin is a substance with a unique smell produced in the process of shredding and squeezing garlic (*Allium sativum*), a species in the onion genus [9]. Allicin is extremely unstable and easily transforms into a series of sulfides under a suitable environment, such as diallyl sulfide, diallyl disulfide (DADS), diallyl trisulfide (DATS), etc. [10]. Since DATS has the highest content of organic sulfur compounds, accounting for 41.5%, it is considered to be the most valuable compound for research in allicin [11,12]. DATS has many medical properties, including cholesterol lowering, blood lipid lowering, and antioxidant and anti-tumor activity [13,14,15]. DATS is a potential chemopreventive substance, which can provide protection against chemical induction in cancer cells [16]. It is a natural donor of hydrogen sulfide (H_2_S) compound, which acts as a gaseous signaling molecule in a variety of physiological processes and is also a therapeutic agent in suppressing oxidative stress and combating cellular apoptosis [17,18]. DATS is considered to be a biological antibacterial agent that has inhibitory effects on *Penicillium* spp., *Botrytis cinerea*, *Pseudomonas aeruginosa*, *Trametes hirsuta*, *Laetiporus sulphureus*, *Klebsiella pneumoniae*, *Campylobacter jejuni*, etc. [19,20,21,22,23,24]. There were some reports that garlic juice was able to reduce disease severity in potato tuber late blight [25], and garlic crude extracts were found to have inhibitive effects on the growth of *P. infestans* [26], while the underlying mechanism of garlic extracts, juice, or its bioingredients against *P. infestans* is very limited. Therefore, in this study, the most valuable compound of garlic extract, DATS, was chosen to detect whether DATS could prevent and control potato late blight, and the mechanisms were discussed.

Research contents of this study mainly include: (1) the effect of DATS on growth and infection ability of *P. infestans*; (2) the induction effect of potato tubers against *P. infestans* under the treatment of DATS; (3) changes in antioxidant enzyme activity of potato tubers with DATS treatment; (4) response and changes of differentially expressed genes (DEGs) and miRNAs (DEMs) in potato tubers with DATS treatment; and (5) regulatory network of potato defense-related DEGs and DEMs regulated by DATS resistance to late blight in potato tuber.

## 2. Results

### 2.1. Effect of DATS on P. infestans and Potato

In vitro Petri dish, 1 μg L^−1^ DATS had no significant effect on the growth of *P. infestans* hyphae, while 100 μg L^−1^ DATS treatment could inhibit its growth significantly (Figure 1A,B). Among all the treatments, the lesion sizes on the detached potato leaves treated with 100 μg L^−1^ DATS were the smallest (Figure 1C), and the disease index of detached potato leaves was significantly reduced along with the increase of DATS concentration (Figure 1D). The layer of *P. infestans* was thin without obvious infection on the potato cubes treated with 100 μg L^−1^ DATS, while it was clearly visible on the control treatment (Figure 1E). In addition, analysis of the lesion diameter on potato cubes showed that DATS treatment significantly reduced the infection of *P. infestans* (Figure 1F). After pre-treatment of potato tubers with DATS, the spread of the lesion was slow and there were few *P. infestans* hyphae on the potato, while with the control, the lesions on the potato pieces continued to expand over time and there was a clear white hyphae layer from the inoculation wound to the lenticels (Figure 1G,H). These results indicated that DATS not only significantly inhibited the hyphae growth and pathogenicity of *P. infestans* in vitro, but also significantly induced the overall resistance of potato against *P. infestans*.

### 2.2. Effects of DATS Treatment on Antioxidant Enzyme Activity (POD, CAT, SOD) and MDA Content of Potato Tubers

The activities of three antioxidant enzymes (peroxidase (POD), catalase (CAT), superoxide dismutase (SOD)) and malondialdehyde (MDA) content were measured in the potato tubers with or without DATS treatment (Figure 2). The POD activity of all treatment increased over time; however, there were no significant differences between them (*p* > 0.05) (Figure 2A). The SOD activity of DATS and the control treatment decreased in the first 3 d and then that of DATS treatment increased faster than the control on 3 to 8 d (*p* > 0.05) (Figure 2B). The CAT activity of DATS treatment showed an upward trend, while the control was opposite. The CAT activity of DATS treatment was always higher than that of the control, and the difference was significant on 8th day after treatment (*p* < 0.05) (Figure 2C). The MDA content in the DATS treatment was always lower than that in the control (*p* > 0.05) (Figure 2D). These results indicated that DATS had a strong effect on the CAT of potato tubers, while there was no significant effect on other antioxidant enzymes (POD, SOD) and MDA content in potato tubers.

### 2.3. Overview of mRNA Sequencing

The transcriptome sequencing of potato pretreated with or without DATS treatment infected by *P. infestans* obtained 42.99 GB clean data and the average Q30 base percentage was 93.7% (Appendix A). The clean reads were compared with the potato reference genome (http://plants.ensembl.org/Solanum_tuberosum; accessed on 12 July 2019). According to their expression levels between DATS treatment and control, 607 significantly DEGs were observed (|log_2_ FC| ≥ 1 and FDR < 0.01), including 255 significantly upregulated and 352 significantly downregulated genes (Figure 3A, Appendix A). In order to verify the accuracy and reliability of the mRNA transcriptome data, we randomly selected five DEGs for qRT-PCR and the primers were listed in Appendix A. The results showed that the trend of gene expression between mRNA transcriptome and qRT-PCR was very consistent and there was a significant positive correlation between the mRNA transcriptome and qRT-PCR (R^2^ = 0.71, *p* value < 0.0001) (Figure 3B,C). It was found that the GO terms were uniformly assigned to 4282 BP (biological process), 2219 MF (molecular function), and 779 CC (cellular components) categories (Appendix A). GO enrichment analysis of DEGs revealed that the top significantly enriched terms include protein disulfide oxidoreductase activity, heme binding, the integral component of membrane, the term of negative regulation of endopeptidase activity, etc. (Appendix A). According to the KEGG database, the annotated DEGs were divided into five parts, including 80 enriched KEGG pathways (Figure 3D, Appendix A), among which starch and sucrose metabolism, plant hormone signal transduction, phenylpropane biosynthesis, and plant-pathogen interaction were the most enriched pathways.

#### 2.3.1. The Regulatory Effects of DATS Treatment on Transcription Factors

In the study, 48 genes belonging to 27 TF families were significantly differentially expressed. The MYB TF family had the most DEGs (12), followed by AUX/IAA (4), AP2-ERF (4); DEGs were also detected in other TF families like ATHB, WRKY, MYC, HSF, bZIP (Table 1). These genes were mainly involved in plant hormone signal transduction and defense metabolism pathways by KEGG and GO analysis, respectively.

#### 2.3.2. The Regulatory Effects of DATS Treatment on Plant Hormone Signals

The KEGG analysis showed that DEGs were enriched in plant hormone signal pathways, including ET (ethylene), JA (jasmonic acid), and IAA pathways. In the ET signal pathway, the ethylene-responsive transcription factor 1 (ERF1) gene was significantly upregulated. In the JA signal pathway, the genes associated with jasmonic acid-amido synthetase 1 (JAR1), jasmonate ZIM-domain (JAZ), and the basic-helix-loop-helix transcription factor (MYC2) were all significantly downregulated (Figure 4A). In the IAA signal pathway, a transport inhibitor response 1 (TIR1) gene was significantly downregulated, which alleviated the inhibition of three auxin-responsive protein (AUX/IAA) genes, and enabled the downstream auxin-responsive promoter GH3 genes to be significantly differentially expressed (Figure 4B).

#### 2.3.3. The Regulatory Effects of DATS Treatment on ROS

DATS treatment did not significantly affect potato peroxisomes through the activities of POD, SOD, etc. (Figure 2), but activated the expression of NADPH oxidase. The upstream gene calcium-dependent protein kinase (CDPK) of NADPH oxidase was significantly downregulated (Figure 4C).

### 2.4. Overview of miRNA Sequencing

For further investigating the miRNA’s expression in the potato after the treatment of DATS, the miRNA sequencing was conducted on an Illumina Hiseq 2500/2000 platform. A total of 81.20 M clean reads were obtained, and each sample was more than 9.99 M (Appendix A). The number of miRNAs were 451, including 155 known and 296 novel miRNAs (Appendix A). Comparing DATS-treated potato tubers with the control, 60 significantly differentially expressed miRNA (DEMs) (|log_2_(FC)| ≥ 0.5; *p* value ≤ 0.05) were identified, including 36 upregulated and 24 downregulated miRNAs. There were 1487 target genes predicted by the 60 DEMs (Appendix A).

### 2.5. Regulatory Network from the Integrated Analysis of miRNA-mRNA Data

Based on the software Cytoscape (3.7.0), the intersection of 607 DEGs and 1487 target genes of DEMs was used to obtain a miRNA-mRNA co-expressed gene set (Figure 5A), which contains 21 miRNA-mRNA negatively regulated interaction pairs (Figure 5B, Appendix A). COG functional classification analysis showed that miRNA-mRNA co-expressed genes were annotated into 11 sub-functions, mainly concentrated in three sub-functions: K: transcription, M: cell wall/membrane/envelope biogenesis, O: posttranslational modification, protein turnover, chaperones (Figure 5C). KEGG classification analysis showed that miRNA-mRNA co-expression genes were enriched in 12 pathways, including metabolic pathways, biosynthesis of secondary metabolites, starch and sucrose metabolism, etc. (Figure 5D).

### 2.6. Model of DATS Regulating Potato Tuber Late Blight Resistance

Combining the data of phenotype, transcriptome, miRNA, and antioxidant enzyme activity, the possible model of DATS participating in the prevention and control of potato late blight is established (Figure 6). DATS can directly inhibit the mycelial growth of *P. infestans* and induce potato tubers’ resistance to *P. infestans*. The possible induction mechanism includes disease signal perception, plant hormone signal transduction, and transcription factor regulation. Specifically, DATS obtains a certain number of ROS by activating NADPH oxidase, which acts as a signal molecule to further stimulate a variety of defense changes, e.g., plant hormone signal transduction and defense-related TFs, MYB, AP2/ERF, MYC2, and others. In addition, DATS regulates hormone pathway transcription factors and defense-related genes by affecting the expression of miRNA.

## 3. Discussion

Biocontrol based on the use of natural bioactive compounds displays great potential in the suppression of plant diseases. For example, the crude ethanolic extract of *Ageratum conyzoides* L. had anti-oomycete activity against *Phytophthora megakarya* [27]. *Cupressus sempervirens* essential oils successfully control postharvest grey mould disease of tomato by inhibiting the *B. cinerea* infection [28]. In this study, we found DATS from garlic had a good effect on inhibiting the growth of *P. infestans* directly and inducing potato tubers to resist *P. infestans*. DATS also has effective control of postharvest disease *Penicillium expansum* of citrus, the destructive stored-product pest (*Sitotroga cerealella*), the important foodborne pathogen (*Campylobacter jejuni*), etc. [19,23,29,30].

Plant cells can sense microbial-specific molecules, such as pathogen-related molecular patterns (PAMPs), to activate defense responses, including ethylene production, oxidation burst, plant cell wall modification, etc. [31]. It is generally believed that the resistance level of the plant is affected by system signals mediated by plant hormones, especially salicylic acid (SA), jasmonic acid, and ethylene [32,33]. The KEGG analysis in this study showed that DEGs were enriched in JA, ET, and IAA pathways, but not in SA pathways. We speculate that DATS processing is mainly through the key transcription factors that affect the ET and JA pathways (such as MYC2, ERF1) to achieve the regulation of defense genes. DEGs analysis showed that DATS treatment significantly upregulated putative late blight resistance protein homolog R1B-12 (PGSC0003DMG400001983) (Appendix A).

ERF1 encodes a key transcription factor in the ethylene pathway that regulates the expression of pathogen response genes that prevent disease progression [34]. Overexpression of ERF1 enhances plant resistance to necrotizing *Botrytis cinerea* and *Cucumber Fusarium* wilt [35]. In this study, after DATS treatment, ERF1 was significantly upregulated, suggesting ERF1 might participate in the potato defense system to resist *P. infestans*.

The jasmonic acid signal pathway was actually significantly affected by DATS, and JAR1, JAZ, and MYC2 genes were all significantly downregulated. JA regulated the resistance to pathogen infections and insect attacks by triggering genome-wide transcription reprogramming in plants. Transcription factor (TF) MYC2 was the central regulator of the JA signaling pathway [36], which acted downstream of JA receptors [37]. MYC2 positively regulates insect defense, wound response, flavonoid metabolism, and oxidative stress tolerance, and negatively regulates pathogen defense and secondary metabolism [36,38]. From this point of view, DATS significantly reduced the expression of MYC2 transcription factor in the jasmonic acid pathway, which may indirectly enhance the potato’s defense against *P. infestans*. The negative regulation of DATS on the entire jasmonic acid pathway led us to speculate that DATS has great potential as an inhibitor of the jasmonic acid pathway, which requires further experiments to confirm.

In the process of interaction between plants and pathogens, ROS played an important role [39]. To some extent, the various functions of ROS signal transduction were attributed to differences in the regulatory mechanisms of NADPH oxidase. The activation of NADPH oxidase was the first process after MAMP’s sense [40]. When these NADPH genes were silenced by virus-induced gene silencing, the suppressed plants showed a reduction in ROS accumulation, along with a loss in resistance to *P. infestans* [41,42]. In this study, two differentially expressed NADPH oxidases were detected and both were upregulated, but their upstream gene CDPK was downregulated in the calcium-dependent protein kinase pathway (Figure 4). We speculated that DATS did not promote the accumulation of ROS through the calcium-dependent protein kinase pathway. It may directly target NADPH oxidases or indirectly stimulate the expression of NADPH oxidases through other ways. The enzyme activity experiments found that POD and SOD showed no significant difference in expression after DATS treatment. These antioxidant enzymes as ROS scavengers were positively related to ROS accumulation. These results indicate that NADPH stimulated by DATS may be the reason to accumulate ROS as signal molecules to enhance potato resistance to *P. infestans*.

MiRNAs are a class of non-coding RNAs regulating gene expression and play roles in plant development and disease resistance [43,44]. Many transcription factors, such as MYB, ARF, HD-Zip, and AP2, are the targets of miRNAs [45,46], suggesting that miRNAs are located in the core position of the network in gene regulation [44]. The expression and the variation of miR482 and miR2118 reflected the shift in the balance of plants’ NBS-LRR defense systems [47]. In this study, 21 potential miRNA/targets pairs were identified through a combined analysis of the datasets of miRNA sequences and mRNA expression profiling. Among them, transcription factor and putative late blight resistance protein might be regulated by novel_miR_84, novel_miR_311, novel_miR_218, and novel_miR_140. This indicates that DATS treatment may regulate the defense response of potato to *P. infestans* by affecting the expression of miRNAs. The specific functions and regulatory networks of these novel miRNAs in potato disease resistance have not been reported yet. These novel miRNAs may be new breakthroughs in the control of *P. infestans* in the future, and they are worthy of more in-depth research.

## 4. Materials and Methods

### 4.1. Culture Medium, Strain, Plant Material

The *P. infestans* strain used in this study is 88069 (Pi88069), A2 mating type. DATS (HPLC ≥ 98%) was purchased from Chengdu Purechem-Standard Co., LTD, China. The rye medium (RSA) was used for the cultivation of *P. infestans* [48]. The potato variety was Jizhang No. 12, which was highly sensitive to *P. infestans*.

### 4.2. The Effect of DATS on the Growth of P. infestans In Vitro

DATS was dissolved in diethyl ether solution and prepared in 15 mg·mL^−1^ mother liquor. The mother liquor DATS was added to RSA, which was cooled to about 55 °C after high temperature sterilization of 121 °C for 20 min, to make 1, 10, and 100 μg L^−1^ plates. RAS without DATS was set as the control and three replicates were set up. The center of the plates was inoculated with *P. infestans* (7 mm diameter, 10 d old), and then cultured in the dark at 20 °C for 9 d. The colony diameter was measured by a ruler with the cross method [49] and photographed every day.

### 4.3. The Effect of DATS on the Infection of P. infestans Sporangia on the Detached Potato Leaves and Cubes

Preparation of sporangia suspension: wash Pi88069 spores (10 d old) with sterile water, and filter it through a layer of sterile magic filter cloth. The sporangia were counted using a hemocytometer and the suspension was adjusted to 5 × 10^4^ sporangia mL^−1^ with sterile water. The mother liquor DATS was added to make sporangia suspensions containing 0, 50, 100 μg L^−1^ DATS.

Potato leaves with the same leaf age and without disease were picked and washed with sterile water (15 replicates per treatment) [50]. The potato tubers without disease were peeled and cut into cubes with 6 × 4 × 0.5 cm in length, width, and height (5 replicates per treatment). The sterile filter paper was laid on a rectangular stainless-steel dish (1 m × 0.5 m). A total of 20 μL fresh configured sporangia suspension containing 50 and 100 μg L^−1^ DATS were added to the center of the leaf and the sporangia suspension containing 100 μg L^−1^ DATS was added in the center of the potato cubes. After the sporangia suspension dried, the entire rectangular stainless-steel dish was covered with plastic wrap to keep it moist under 90% humidity. For the cubes, the dishes were incubated at 20 °C for 5 d in a dark environment, and for the leaves, in a 12 h/12 h light/dark cycle. The lesion on the potato cubes was photographed and measured by ImageJ. Potato late blight in detached leaves was assessed statistically based on grading standards (Appendix A). The disease index was calculated as follows:$$\mathrm{Disease}\;\mathrm{index}\;\left(\%\right)=\sum \frac{\;\mathrm{number}\;\mathrm{of}\;\mathrm{lesions}\times \mathrm{number}\;\mathrm{of}\;\mathrm{disease}\;\mathrm{leaves}}{\mathrm{total}\;\mathrm{number}\;\mathrm{of}\;\mathrm{leaves}\times \mathrm{highest}\;\mathrm{number}}\times 100$$

### 4.4. The Effect of DATS on Improving the Resistance of Potato Tubers against P. infestans

Potato tubers, without visible defects or rot, were cleaned up with water and sanitized in 75% alcohol and 2% sodium hypochlorite for 2 min, respectively. The sanitized tubers were rinsed with sterile distilled water three times and air-dried after soaking in 100 μg L^−1^ DATS solution or sterile water (CK) for 20 min. They were washed with sterile water. After 24 h, we used a sterile punch with a diameter of 6 mm to make two holes (8 mm depth) in the potato tubers, where 20 μL *P. infestans* sporangia suspension were inoculated. The sporangia suspension was made in the same method as in Section 4.3. The potato tubers were put on the stainless-steel square plate, which was covered with plastic wrap to keep the moisture. The potato tubers were taken out on the 3rd, 5th, and 8th day and cut longitudinally along the line of two holes. The lesion diameter was measured near the holes with a ruler. The mycelium was thrown away and 1 to 3 cm tuber tissues around the hole were quickly obtained and frozen with liquid nitrogen, and stored at −80 °C for mRNA and miRNA sequencing and determination of physiological and biochemical indicators. A total of 72 potato tubers were used, among which there were 36 for treatment and 36 for CK. Each experiment had three replicates.

### 4.5. Antioxidant Enzyme Activity Detection

The preserved tuber tissue in Section 4.4 was ground with liquid nitrogen, and each sample had 0.1g. Enzyme activity determination was carried out according to the instructions of a POD, SOD, CAT, MDA kit purchased from Nanjing Jiancheng Institute of Biological Engineering, China. The instructions for the kits related to enzyme activity can be found at the following website: http://www.njjcbio.com/ (accessed on 17 March 2019).

### 4.6. MRNA and miRNA Transcriptome Sequencing

The 3rd day potato tuber tissues in Section 4.4 were used for miRNA and mRNA transcriptome sequencing. An RNAprep Pure Plant Total RNA Isolation Kit (Tiangen Biochemical Technology Co., Ltd, Beijing, China) was used to isolate the total RNA. The integrity of the total RNA was checked by 1% agarose gel electrophoresis, and the concentration and purity were determined with a Nanodrop™ 2000 spectrophotometer (Thermo Scientific, Waltham, MA, USA). MiRNA and mRNA transcriptome sequencing were completed by Biomarker Technologies (Beijing, China).

MRNA sequencing: based on Sequencing by Synthesis (SBS) technology, the Illumina high-throughput sequencing platform was used for constructing cDNA libraries. The DEGs between two samples were determined based on a |log_2_ FC| ≥ 1 and a false discovery rate (FDR) < 0.01. The heat map tool in BMKCloud platform (https://international.biocloud.net/zh/software/tools/list; accessed on 18 January 2020) was used to cluster the DEGs. GO and KEGG analysis were performed with GO and KEGG classification enrichment map tools (https://www.geneontology.org/ and https://www.genome.jp/kegg/; accessed on 19 January 2020).

MiRNAs sequencing: the NEBNext Multiplex Small RNA Library Prep Set for Illumina (NEB, USA) was used to generate the sequencing libraries, whose quality was assessed by the Agilent Bioanalyzer 2100 system. The clustering of the index-coded samples was performed on a cBot Cluster Generation System using TruSeq PE Cluster Kit v4-cBot-HS (Illumia). The library preparations were sequenced on an Illumina platform and single-end reads were generated. Differential expression analysis of two groups was performed using the DESeq2 (1.10.1) R package [51]. MiRNAs with |log_2_ Fold Change (FC)| ≥ 0.50; *p* value ≤ 0.05 were assigned as differentially expressed. The TargetFinder software was used to predict target genes of DEMs in potato [52]. 

The software Cytoscape (3.7.0) was used to construct a miRNA-mRNA co-expression regulatory network.

### 4.7. Real Time-Quantitative PCR

The same total RNA used in the transcriptome sequencing was used here. Primer Premier 5.0 software was used to design real-time quantitative primers (Appendix A). The reaction was carried out in a final volume of 25 μL, which contained 12.5 μL 2 × SYBR^®^Premix Ex Taq (TAKARA, Dalian, China). The relative expression level of each target gene was analyzed with Bio-Rad CFX Manager software. The data represented the mean ± standard error (SE) of 3 independent experiments.

### 4.8. Statistical Analysis

The one-way ANOVA process of the software SAS v9.0 was used for the analysis of variance. The data were tested for normality and homogeneity of variance, and they can satisfy the assumptions of normality and homoscedasticity for ANOVA. The software GraPhpad 5.0 was used for mapping.

## 5. Conclusions

In this study, DATS can improve the potato’s biocontrol ability against *P. infestans* and the possible mechanisms suggest that: (1) DATS inhibits the growth of *P. infestans* hyphae directly and reduces the pathogenicity of *P. infestans* on detached leaves and tubers; (2) DATS induces the overall resistance of potato against late blight; (3) DATS affects the level of CAT significantly and has no significant effect on other antioxidant enzymes (POD, SOD) and MDA in potato tubers; (4) GO an KEGG analysis shows that DEGs are mainly involved in ROS biogenesis, plant hormone signal transduction, and defense-metabolism-related pathways; (5) twenty-one miRNA-mRNA negatively regulated interaction pairs were detected and some key DEMs target the hormone pathway transcription factors and defense-related DEGs. In summary, this study provides a new idea for the development of DATs as an eco-friendly agent for the control of potato late blight.

## Figures and Tables

**Figure 1 ijms-24-03474-f001:**
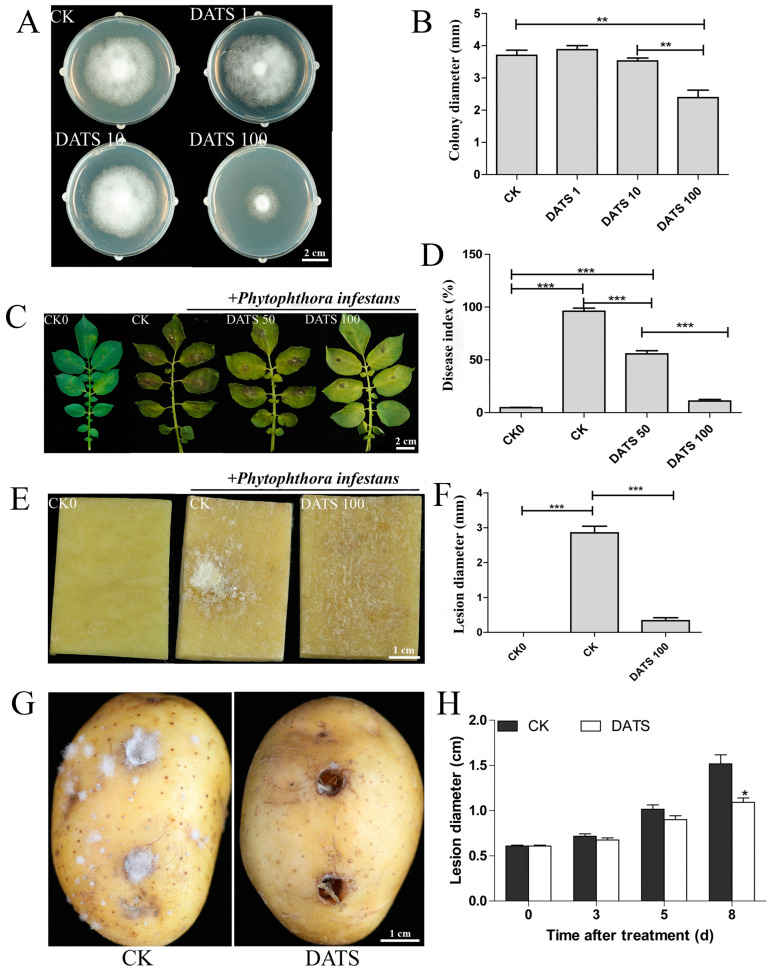
Effect of DATS on *P. infestans* and potato. (**A**) Effect of DATS on the growth of *P. infestans* mycelium. (**B**) The colony diameter of *P. infestans* treated by DATS. (**C**) Effect of DATS on *P. infestans* on the potato leaves. (**D**) Disease index of potato leaves infected by *P. infestans* with DATS. (**E**) Effect of DATS on *P. infestans* on the potato cubes. (**F**) Lesion diameter of potato cubes infected by *P. infestans* with DATS. (**G**) The potato tubers pretreated by DATS (100 μg L^−1^) infected by *P. infestans*. (**H**) Lesion diameter of the potato tubers pretreated by DATS infected by *P. infestans*. LSD was calculated to compare significant differences between DATS treatment and CK treatment, * *p* < 0.05, ** *p* < 0.01, *** *p* < 0.001, unmarked = not significant, n ≥ 3. Data are shown as means ± standard deviation (SD). CK0 indicates neither *P. infestans* infection nor treatment with DATS; CK indicates the detached leaf or potato tuber is not treated with DATS; DATS1, 10, 50, 100 indicate the concentration of DATS (1, 10, 50, 100 μg L^−1^); DATS: diallyl trisulfide.

**Figure 2 ijms-24-03474-f002:**
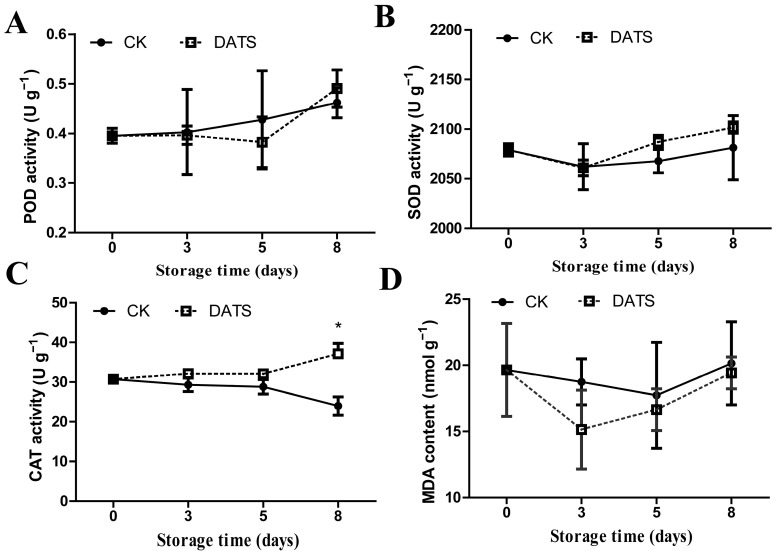
Effect of DATS treatment on enzyme activities of POD (**A**), SOD (**B**), CAT (**C**), and MDA (**D**) of potato tubers. CK: control; DATS: diallyl trisulfide; POD: peroxidase; SOD: superoxide dismutase; CAT: catalase; MDA: malondialdehyde. * *p* < 0.05; data are shown as means ± SD.

**Figure 3 ijms-24-03474-f003:**
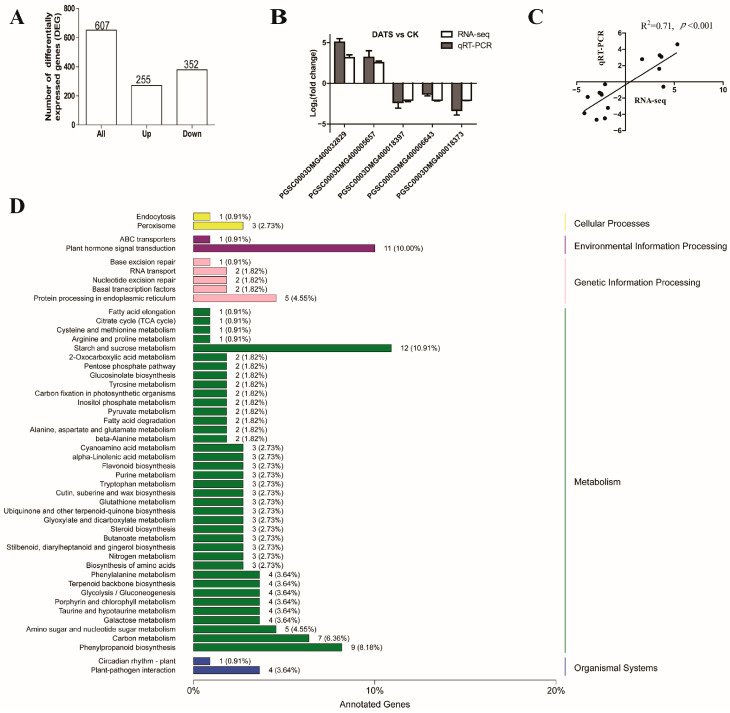
The basic information of the RNA transcriptome and qRT-PCR data analysis after DATS treatment. (**A**) Upregulated and downregulated genes; (**B**) verification of expression pattern of DEGs by qRT-PCR. (**C**) Correlation analysis between mRNA transcriptome and qRT-PCR data. (**D**) Classification map of DEGs in KEGG; the ordinate is the name of the KEGG metabolic pathway, and the abscissa is the number of genes annotated to the pathway and its proportion to the total number of genes annotated. Data were shown as means ± SD.

**Figure 4 ijms-24-03474-f004:**
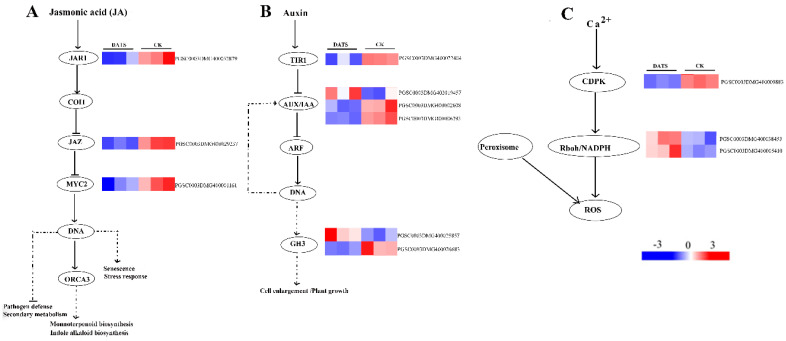
Effects of DATS treatment on the regulation of jasmonic acid signal pathway (**A**), auxin signal pathway (**B**), and ROS biosynthesis (**C**).

**Figure 5 ijms-24-03474-f005:**
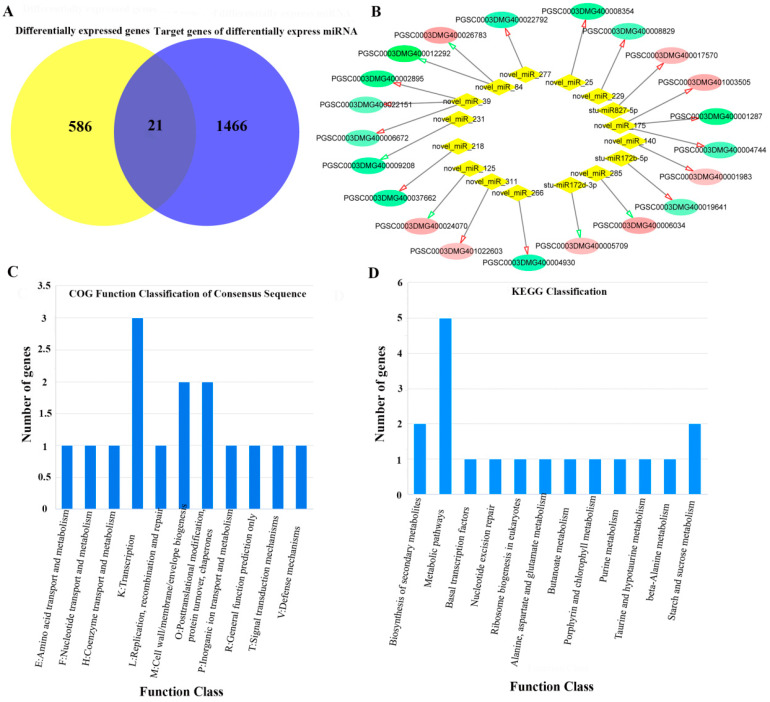
Regulatory network from the integrated analysis of miRNA-mRNA data. (**A**) The intersection between DEGs and target genes of DEMs. (**B**) The regulatory network between DEGs and target genes of DEMs. Note: The red circle represents upregulated DEGs and the green circle represents downregulated DEGs. Red arrow indicates upregulated miRNA, green arrow indicates downregulated miRNA. (**C**) COG function classification based on the intersection between DEGs and target genes of DEMs. (**D**) KEGG function classification based on the intersection between DEGs and target genes of DEMs.

**Figure 6 ijms-24-03474-f006:**
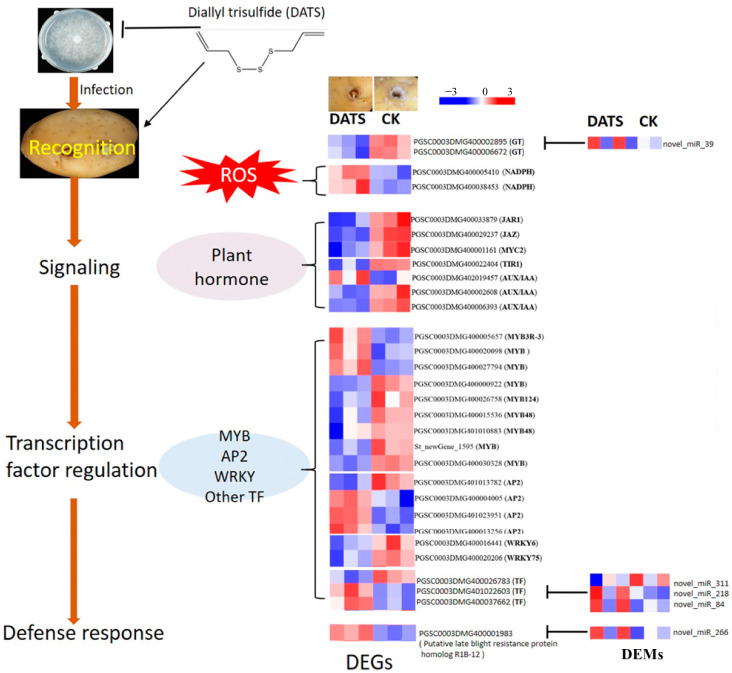
Model of DATS regulating the resistance of potato to *P. infestans*.

**Table 1 ijms-24-03474-t001:** DEGs belonging to transcription factor (*p* < 0.05).

#ID	log_2_FC	Regulated	FDR	Pfam_Annotation
PGSC0003DMG400003568	1.11	Up	2.78 × 10^−5^	Zinc knuckle
PGSC0003DMG400016441	−1.23	Down	0.001005508	WRKY DNA-binding domain
PGSC0003DMG400020206	−1.29	Down	0.001670048	WRKY DNA-binding domain
PGSC0003DMG400023462	1.11	Up	0.007290252	Ubiquitin family
PGSC0003DMG400002052	1.07	Up	5.82 × 10^−8^	Transcription initiation factor IIA
PGSC0003DMG400006588	1.52	Up	1.38 × 10^−7^	Timeless protein C terminal region
Solanum_tuberosum_newGene_4204	−1.37	Down	5.34 × 10^−6^	TCP family transcription factor
PGSC0003DMG400027770	1.16	Up	0.003989554	Homeobox domain
PGSC0003DMG400003846	1.03	Up	0.009284718	RNA polymerase Rpc34 subunit
PGSC0003DMG400009063	1.19	Up	2.80 × 10^−8^	Ribosomal proteins L26 eukaryotic
Solanum_tuberosum_newGene_1998	1.24	Up	0.005085243	Regulator of chromosome condensation (RCC1) repeat
PGSC0003DMG400026195	1.38	Up	3.13 × 10^−6^	Protein of unknown function, DUF573
PGSC0003DMG400021211	1.96	Up	2.87 × 10^−7^	Piwi domain
PGSC0003DMG400014182	1.01	Up	0.002884298	NF-X1 type zinc finger
PGSC0003DMG400003055	1.53	Up	5.67 × 10^−7^	Myb-like DNA-binding domain
PGSC0003DMG400026758	−1.10	Down	0.000566068	Myb-like DNA-binding domain
PGSC0003DMG400015536	−1.13	Down	0.003151643	Myb-like DNA-binding domain
PGSC0003DMG400005657	2.55	Up	2.74 × 10^−12^	Myb-like DNA-binding domain
PGSC0003DMG401010883	−1.41	Down	0.004240926	Myb-like DNA-binding domain
PGSC0003DMG400030328	−1.58	Down	1.33 × 10^−11^	Myb-like DNA-binding domain
PGSC0003DMG400027794	1.13	Up	0.000647544	Myb-like DNA-binding domain
PGSC0003DMG400000922	−1.04	Down	0.000123933	Myb-like DNA-binding domain
PGSC0003DMG401024549	−1.44	Down	3.33 × 10^−7^	Myb-like DNA-binding domain
Solanum_tuberosum_newGene_4889	−1.09	Down	1.25 × 10^−5^	MYB-CC type transfactor
Solanum_tuberosum_newGene_1595	−1.52	Down	2.40 × 10^−5^	Myb/SANT-like DNA-binding domain
PGSC0003DMG400020098	1.15	Up	0.000702071	Myb/SANT-like DNA-binding domain
PGSC0003DMG400014811	1.00	Up	0.001674446	HSF-type DNA-binding
PGSC0003DMG400025342	−1.18	Down	0.000950719	Homeobox domain
PGSC0003DMG400016148	−1.51	Down	4.77 × 10^−5^	Homeobox domain
PGSC0003DMG400018509	−1.15	Down	0.000430995	Homeobox domain
PGSC0003DMG400006132	−1.02	Down	0.00568511	GRAS domain family
PGSC0003DMG400025001	2.10	Up	2.60 × 10^−12^	FACT complex subunit (SPT16/CDC68)
PGSC0003DMG400026783	1.73	Up	4.24 × 10^−7^	ERCC3/RAD25/XPB C-terminal helicase
PGSC0003DMG400002507	−1.22	Down	0.003764879	Dof domain, zinc finger
PGSC0003DMG400010684	1.10	Up	0.004424802	GATA zinc finger
PGSC0003DMG401010056	−1.92	Down	2.81 × 10^−10^	CCT motif; B-box zinc finger
PGSC0003DMG400000272	1.10	Up	0.007616122	CCAAT-binding transcription factor (CBF-B/NF-YA) subunit B
PGSC0003DMG400016383	1.04	Up	0.007395023	CAF1 family ribonuclease
PGSC0003DMG400030946	−1.35	Down	0.002894626	bZIP transcription factor
PGSC0003DMG400001161	−1.29	Down	0.006448084	bHLH-MYC and R2R3-MYB transcription factors N-terminal
PGSC0003DMG402019457	1.15	Up	0.006582071	AUX/IAA family
PGSC0003DMG400006393	−1.28	Down	2.26 × 10^−10^	AUX/IAA family
PGSC0003DMG400002608	−1.51	Down	0.000393116	AUX/IAA family
PGSC0003DMG400019635	−1.01	Down	0.004468473	Associated with HOX
PGSC0003DMG400013256	1.45	Up	0.000291407	AP2 domain
PGSC0003DMG400004005	1.09	Up	0.004163584	AP2 domain
PGSC0003DMG401023951	1.13	Up	1.71 × 10^−6^	AP2 domain
PGSC0003DMG401013782	−1.31	Up	3.89 × 10^−5^	AP2 domain

## Data Availability

The datasets generated during and/or analysed during the current study are available from the corresponding author on reasonable request.

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
