# Peer review of "Integrative mRNA and microRNA Analysis Exploring the Inducing Effect and Mechanism of Diallyl Trisulfide (DATS) on Potato against Late Blight"

_ijms, 2023, doi:10.3390/ijms24043474_

Round 1
Reviewer 1 Report
Over all comments: Thanks for reporting on this work it is interesting, and you have pulled out some great results. Suggest being more succinct in materials and methods section. Avoid using things such as ‘the, and, and then, at last’ at the beginning of a sentence. There needs to be more detail in the methods so that it can be repeated by another researcher for instance in line 341 you mention the tubers were ‘inoculated with P. infestans’ but no detail is given about the inoculum. more replicates are needed especially for potato tubers in 4.3 4.5 and 4.6. Uncertain why all the concentrations of DATS where not used for all the experiments, this would have rounded things off nicely Due to the small number of replicates and the fact that this is an in vitro rather than in vivo study the results are an indication, and the conclusions should be written as such. Line 403 and 404 are an excellent example of how the conclusion should be written. I have not made much comment on the results section, but I will note that this section should be just results. No references should be needed in your results. Draw conclusions or give explanations in the conclusions/discussion portion of the manuscript.
Below are some changes, note I have not gone through the results for reasons I have given in the general comments.
Line 14, 21 etc: Biological control should be replaced with biocontrol. Reference to control of a disease using a natural substance (biocontrol) not a living organism (biological control).
Line 20-22: needs rewording as it is confusing.
Line 27: Key words should not be in the title.
Line 30: to be consistent the Latin name should be used for potato after the common name when referring to it for the first time in the text.
Line 34: remove the ‘ after P. infestans. Replace spores with zoospores since it also has caducous sporangia this is also a method of dispersal?
Line 41: wrong use of the word ‘drug’ suggest ‘chemical’
Line 57: for consistency use Penicillium spp. or species
Line 58: lower case j in ‘Jejuni’
Line 63-64: Suggest ‘the most valuable compound of garlic extract, DATS, was chosen to detect whether they could prevent and control potato late blight, and the mechanisms are discussed.’
Line 306-309: Jiasui Zhan should be acknowledged in the acknowledgements. Unclear if this strain has been lodged in a registered collection if not it needs to be and it needs to be identified correctly with the acronym of the registered collection. Consider including the recipe or a reference for the rye medium.
Line 311-315: how was the DATS added to the plates? What size inoculum was used? Was only one diameter taken for each plate? I would suggest two measurements at right angles which are then averaged and then the average of your three replicates be converted to mean radial growth day-1 and graphed. Why was DATS 50 not used? Suggest replacing ‘hyphae’ with ‘in vitro’ in methods and results. ‘3 replicates’ should be ‘three’.
Line 318-333: Remove ‘The’ and ‘At last’ from the beginning of sentences. The preparation of sporangial suspension should be in order of how the experiment was carried out. This section could be worded better and ensure that you include the amount of time between adding the DATS to the sporangial suspensions and inoculation of plant material.
Line 331-332: suggest ‘using a hemocytometer’
319: ‘disease’ not ‘diseases’
Line 346: remove ‘them’
Line 347: ‘had three replicates’
Best of luck with your manuscript, I look forward to reading the published version.
Reviewer 2 Report
This manuscript presents a study evaluating a potential biological control agent, DATS, for late blight, a major potato disease. The authors demonstrated that DATS has positive effects on the activation of the potato defense system. By checking the activity changes of related genes and enzymes, the authors concluded that DATS has the function of inhibiting the pathogen that causes potato late blight. I think the study is original and sound, and the manuscript is well-written. I recommend that it be accepted with minor revisions.
Line 16: Suggest changing to: “This study shows that DATS can …” or “In this study, DATS was found to be able to inhibit …”
Line 38: Suggest changing to: “were used in a large amount.”
